# Zinc and Provitamin A Biofortified Maize Genotypes Exhibited Potent to Reduce Hidden—Hunger in Nepal

**DOI:** 10.3390/plants11212898

**Published:** 2022-10-28

**Authors:** Krishna Dhakal, Amar Bahadur Pun Magar, Keshab Raj Pokhrel, Bandhu Raj Baral, Abdurahman Beshir, Hari Kumar Shrestha, Shree Prasad Vista

**Affiliations:** 1Nepal Agricultural Research Council, Directorate of Agricultural Research Karnali Province, Dasharathpur, Surkhet 21702, Nepal; 2Nepal Agricultural Research Council, Hill Crops Research Program, Dolakha 45500, Nepal; 3International Maize and Wheat Improvement Centre (CIMMYT), Khumaltar, Lalitpur 44705, Nepal; 4Nepal Agricultural Research Council, National Soil Science Research Centre, Khumaltar, Lalitpur 44705, Nepal

**Keywords:** zinc, provitamin A, three-way cross maize, hidden hunger, multi-stress tolerance

## Abstract

Zinc deficiency affects one third of the population worldwide, and vitamin A deficiency is a prevalent public health issue in Sub-Saharan Africa and South-Asia, including Nepal. Crop biofortification is the sustainable solution to these health—related problems, thus we conducted two different field trials in an alpha lattice design to identify zinc and provitamin A biofortified maize genotypes consistent and competitive in performance over the contrasting seasons (Season 1: 18 February to 6 July 2020 and Season 2: 31 August to 1 February, 2020/21). In our study, the performance of introduced maize genotypes (zinc—15 and provitamin A biofortified—24) were compared with that of the local check, focusing on the overall agro-morphology, yield attributes, yield, and kernel zinc and total carotenoid content. Zinc and total carotenoid in the tested genotypes were found in the range between 14.2 and 24.8 mg kg^−1^ and between 1.8 and 3.6 mg 100 g^−1^. Genotypes A1831-8 from zinc and EEPVAH-46 from provitamin A biofortified maize trial recorded kernel zinc and total carotenoid as high as 52.3, and 79.5%, respectively, compared to the local check (DMH849). The provitamin A genotypes EEPVAH-46 and EEPVAH-51 (total carotenoid: 3.6 and 3.3 mg 100 g^−1^), and zinc biofortified genotypes A1847-10 and A1803-42 (20.4 and 22.4 mg kg^−1^ zinc) were identified as superior genotypes based on their yield consistency over the environments and higher provitamin A and zinc content compared to the check. In addition, farmers can explore August sowing to harvest green cobs during December-January to boost up the emerging green cob business.

## 1. Introduction

Micronutrient deficiency attributes to the global burden of diseases by elevating the instances of illness and mortality from disease infection and mental disabilities [1]. The extent of micronutrient deficiency in South-Asia, including Nepal, is alarming. Nepal reports 32 to 35% and 11.3 to 23.3% stunting and wasting in different age groups due to a low intake of essential micronutrients in the daily diet [2,3]. Zinc deficiency is widespread among children (6 to 59 months—21%) and non-pregnant woman (15 to 49 years—24%), and vitamin A deficiency is 4.2% and 3% respectively for the children and non-pregnant woman [3]. Earlier reports by the World Health Organization indicated that more than 32% of pre—school children had vitamin A deficiency disorders [4]. However, nutrient supplementation programs have helped to reduce vitamin A deficiency in recent years as the program is reported to cover more than 90% of children in Nepal [3,4].

Zinc deficiency affects one third of the population worldwide, and vitamin A deficiency is a prevalent public health issue in Sub-Saharan Africa and South—Asia, including Nepal [5,6,7]. Adult women and men above 19 years require 8 and 11 milligrams per day, and 700 and 900 μg retinol activity equivalents (RAE) per day of zinc and vitamin A, respectively [8]. Insufficient zinc affects growth, weakens the immune system and reproductive health, and reduces duration and severity of infection [9,10,11,12]. Vitamin A deficiency majorly causes eye diseases. It increases maternal and childhood mortality, diarrhea and respiratory diseases, and risk of death from infection [5].

The affordability of dietary diversification, food fortification, and nutrient supplementation in the developing world is quite low, and relies heavily on funding sources [13]. Thus, crop biofortification of staple cereals and legumes is the most sustainable way to eliminate micronutrient deficiency at the large scale. Yellow kernel maize varieties commonly cultivated in farmers’ fields contain less than 2 µg g^−1^ of provitamin A, and the average zinc (baseline value) content is about 20 µg g^−1^ [14]. In Nepal, the National Maize Research Program (NMRP) released the first ever protein biofortified maize, Posilo Makai-1 in 2008 and Posilo Makai-2 in 2018, the latter contains 0.42% Lysine and 0.114% Tryptophan [15]. Later, the Grain Legumes Research Program (GLRP) released the first zinc and iron biofortified lentil variety, Khajura Masuro-3 (average iron and zinc content is 81.5 and 65.2 mg kg^−1^) in 2017 [16]. The commodity program then released Khajura Masuro-4 in 2018. Similarly, five iron and zinc biofortified varieties of wheat (Himganga, Bheriganga, Khumal Shakti, Zinc Gahu-1, and Zinc Gahu-2) were released by the National Wheat Research Program (NWRP) and National Plant Breeding and Genetics Research Centre (NPBRC) in 2021 [15,17]. The iron and zinc content in the kernel of these varieties ranged between 32.3–41 and 34.5–54 mg kg^−1^, respectively [18]. The Nepal Seed and Fertilizer Project initiated the introduction and evaluation of provitamin A and zinc biofortified maize varieties, and some of these are in the pipeline to release or register in the country.

Maize is the staple food crop to the majority of people living in the hilly region of Nepal [19]. Geographical difficulty, poor access and affordability of quality food and health facilities, and widespread micronutrient deficiency in soil have exacerbated the micronutrient deficiency in the rural people [20,21,22]. This is quite evident in Karnali Province where stunting (55%), wasting (8%—children under 5 years), and underweight cases in children (36%) are higher [21,22]. The Nepal national micronutrient status survey 2016 disseminated that the deficiency of vitamin A and zinc is prevalent in rural people living in hills and mountains of mid (Karnali Province) and far—western Nepal (far—western province) [3]. Considering this fact, the present study was undertaken to identify zinc and provitamin A enriched maize genotypes with competitive and consistent yielding ability over the contrasting seasons in the river basin area of Karnali Province, Nepal.

## 2. Materials and Methods

### 2.1. Study Location, Soil, and Weather Details

The Field trials were conducted at Directorate of Agricultural Research, Karnali Province, Dasharathpur, Surkhet, Nepal. The geographical coordinates and elevation of the study site are 28°30′ N, 81°47′ E, and 490 masl, respectively. Analysis of the composite soil sample taken from the experimental site (in different years) indicated that the soil was slightly acidic to neutral (pH—6.11 to 6.45), medium organic matter (1.87 to 2.02%), nitrogen (0.09 to 0.10%), phosphorous (30.57 to 96.12 mg kg^−1^), and low to medium potassium (104.50 to 125.80 mg kg^−1^) with sandy loam texture [19,23,24]. The details of agro-meteorological parameters recorded during two growing seasons are displayed in Figure 1.

### 2.2. Experimental Detail and Genetic Materials

Two field experiments were conducted simultaneously in two contrasting seasons; season 1 (18 February–6 July) and season 2 (31 August–1 February) of 2020/21 in the same block. The first season resembles spring, and the second season had the Summer month; September, and rest of Winter; October to February. In the experiment, one trial set consisted of regional extra early multi—stress tolerant provitamin A biofortified maize genotypes (hereafter EEPVAH trial), whereas the other composed of three-way cross zinc enriched maize genotypes (hereafter TWC trial). The hybrids in both the trials were developed by the International Institute of Tropical Agriculture (IITA), Ibadan, Nigeria and introduced in Nepal by the International Maize and Wheat Improvement Centre (CIMMYT), Kathmandu, Nepal through the Nepal Seed and Fertilizer Project.

The EEPVAH and TWC maize trial consisted of 25 and 16 genotypes, respectively, (Table 1). DMH849 was used as a local check, which is an Indian hybrid widely grown by the farmers around the experimental location. Check-RE (TZEE-Y Pop STR C5 x TZEEI 58) in EEPVAH trial was an extra early provitamin A genotype (11.4 μg g^−1^) released in Ghana, Nigeria and Mali and has 5 t ha ^−1^ yield potential. Alpha lattice design was employed (5 × 5 lattice in EEPVAH, and 4 × 4 lattice in TWC) and genotypes were replicated twice. The experimental plots were composed of two rows of 5 m length, and inter and intra—row spacing of 75 cm and 40 cm (50 cm in TWC trial), respectively. Intra—row spacing (plant to plant) in our experiment was nearly double than the recommended spacing for maize (20 to 25 cm) due to the limited seed available to conduct the trial. Two seeds were placed in each hill during sowing, and later thinning was done to maintain a single stand at the 3–4 leaf stage. Fertilizer doses of 120:60:40 NPK kg ha^−1^ and 8 tons ha^−1^ farm yard manure (FYM) were applied. A half dose of nitrogen, full doses of phosphorus, potassium, and FYM were applied as a basal dose during the final land preparation. The remaining dose of nitrogen was applied in two equal splits at V6 and V10 stages of the crop growth. The government’s protocol for integrated pest management of the fall armyworm was employed to control the fall armyworm during the growing seasons [25].

### 2.3. Data Recording Procedures

We recorded data related to phenology, growth and yield attributing traits, cob characteristics, and grain yield. Days to tasseling and silking were recorded when 50% of plants shed pollen grains, and with the emergence of 2–3 cm long silk in 50% of plants in the plot. Plant and ear heights were recorded in centimeters from five central plants. The distance from plant base to the first tassel branch was recorded as plant height, whereas ear height was measured as a distance between plant bases to the node bearing the uppermost ear. The number of plants with a stalk broken below the ear were counted in each plot and converted into a percentage to record the stalk lodging. At the time of harvest, we recorded the total number of plants and ears harvested in each plot. Cob characteristics; cob length and diameter (cm), number of kernel rows per cob, and kernel number in a row were recorded from five randomly selected cobs in each plot as a post—harvest recording. Grain yield, shelling percentage, harvest index, and hundred kernel weights were recorded after proper drying and shelling of the cobs. The ratios of grain recovered to the total dry cob weight, and total grain weight to the biological yield (grain and stover yield) in the plot were recorded as shelling percentage and harvest index, respectively. The grain yield is reported in tons per hectare by adjusting at 14% moisture content (Wile 55 moisture meter used). One hundred dried kernels in each plot were counted and weighed to determine one hundred kernel weights in grams. The standard data recording procedure was followed for recording all the studied parameters [26].

### 2.4. Kernel Nutrient Analysis (Total Carotenoid and Zinc)

After harvesting, composite grain samples were taken from each genotype of EEPVAH and TWC trials to analyze the total carotenoid and zinc content, respectively. The well—dried grain samples (65 °C for 72 h) were grounded, packed, and then sealed in a polythene bag with respective leveling. Each sample was at least 200 g in weight when submitting to the laboratory (National Food Research Centre, Khumaltar, Nepal). For carotenoid analysis (EEPVAH trial), the pigment present in the grounded sample was extracted in di-acetone alcohol, and later was transferred to petroleum ether. Methanolic KOH was used for saponification of chlorophyll and then washed with water to remove it from the mixture. Finally, the carotenoid amount was estimated through the spectrophotometric method at 450 nm β—carotene standard [27].

The USEPA method 3050B (second revised version of method 3050) was employed to determine zinc content of the grain samples taken from TWC trials [28]. The quantitative amount of the crushed, sieved, and dried plant matter sample was digested with tri-acid solution (nitric acid, hydrochloric acid, and per chloric acid) in a beaker followed by digestion over a controlled hot plate and quantitative filtration into a known volumetric flask. The aliquot was aspirated into the AAS at an air-acetylene flame after blank, and three consecutive working metal standard solutions. The final concentration of the respective element was calculated from the values obtained through the software-displayed values on the basis of the calibration curve.

### 2.5. Statistical Analysis

Recorded raw data were entered and processed in Microsoft Excel 2007, and ANOVA was generated with ADEL-R software version 2.0 [29]. Statistical analysis was performed individually for the two separate seasons in both the trials. We chose the randomized incomplete block design (RIBD) while operating data analysis in ADEL-R. The Fisher least significant difference test (*p* < 0.05) was performed in statistically significant response variables to separate treatment means. In order to identify consistent performers (in terms of grain yield) in both the seasons, AMMI analysis was performed using GEA-R software version 4.1 [30].

## 3. Results

### 3.1. Phenological Traits

The evaluated genotypes were highly significant for the days to tasseling and silking in both the trials and in both the seasons. However, anthesis-silking interval was found non-significant (Table 2 and Table 3). The genotypes took a short time to tassel and silk during season-2 in both TWC and EEPVAH trials compared to season-1 (Table 2 and Table 3). The check (RE) recorded the earliest tasseling (70 days) and silking (73 days) during season-1, and was among the early group during season-2 too (tasseling—51 days, and silking—54 days). It was observed that mean anthesis-silking interval during season-2 (4 days) was longer than that in season-1 (2 days). Among the genotypes of the TWC trial, A1830-14 took the longest time for tasseling (86 and 62 days in season-1 and season-2) and silking (88 and 67 days in season-1 and season-2) in both the seasons (Table 3).

### 3.2. Growth and Yield Attributing Traits

Growth and yield attributing traits viz. plant (PHT), ear height (EHT), number of plant (PHPP), ear harvested (EHPP), shelling percentage (SHELP), harvest index (HI), and hundred kernel weight (HKW) were observed in both the trials. A highly significant (*p* < 0.01) effect was observed in all the growth and yield attributing traits in the EEPVAH trial, while the majority of traits were significantly different (*p* < 0.05) in TWC trial (Table 4 and Table 5). The number of plants harvested (PHPP—during season-1 in EEPVAH trial), and hundred kernel weight (HKW—during season-2 in TWC trial) were found non-significant among the tested genotypes (Table 4 and Table 5). Performance of the EEPVAH and TWC genotypes for growth and yield attributing traits were reduced more during season-2 than in season-1 (Table 4 and Table 5). The fall armyworm infestation during season-2 affected the overall vegetative growth and development of the maize genotypes (Figure 2).

Three-way cross zinc biofortified genotype A1803-37 reported the highest dry matter deposition in the kernel (recorded as HKW) during season-1 (36.5 g) as well as in season-2 (30.7 g). A1830-6 and A1831-3 produced statistically similar dry matter content as in A1803-37 (Table 5). The harvest index was recorded highest in the local check (0.6) during season-1 while A1803-37 had the highest (0.6) dry matter deposition in the kernel during season-2. Shelling percentage in the TWC genotypes ranged between 78.7 and 86.1% (mean—82.3%), and between 73 and 80.5% (mean—77.5%) during season-1 and season-2, respectively (Table 5). The mean number of plant and ears harvested were highest during season-1 than in season-2, and similar observations were found for plant and ear height. Plant stature was tallest in 1803-42 (261 cm), and was recorded shortest in A1830-6 (200 cm), and the local check had 202.5 cm during season-1. The plant (172.4 cm) and ear height (69.9 cm) were observed shortest in A1830-9 during season-2 (Table 5).

Among the EEPVAH genotypes, EEPVAH-44 reported maximum dry matter deposition in the kernel in both seasons. The local check also produced higher dry matter (in the form of HKW) in the kernel (35.8 g and 32.4 g), statistically similar to EEPVAH-44 (Table 4). EEPVAH-8 consistently recorded the highest harvest index in both seasons (0.6 in season-1 and 0.7 in season-2). The shelling percentage in EEPVAH genotypes was higher as compared to TWC genotypes. The mean shelling percentage during season-1 and season-2 were 84.7 and 82.5%, respectively (Table 4). Similarly, mean plant and ear height, the number of plant and ears harvested during season-2 were comparatively lower than in season-1. During season-1, the plant stature observed in EEPVAH genotype was as high as 242 cm, and the mean height was 212.2 cm in the overall genotypes. EEPVAH-50 had the shortest plant height (195.4 cm), and the local check recorded 208.5 and 186.6 cm plant and ear height, respectively during season-1 and season-2. Genotype EPVAH-67 reported the tallest plant and ear height during season-2 (Table 4).

### 3.3. Cob Characteristics

During season-1, EEPVAH and TWC genotypes were found highly significant (*p* < 0.01) for most of the cob characteristics (cob length—CL and diameter—CD, and number of kernel-rows per cob—NOKRC) except for the number of kernels per row—NOKPR (Table 6 and Table 7). The EEPVAH genotypes were significantly different in cob characteristics during season-2, however a non-significant difference was observed for CD and NOKRC in TWC genotypes (Table 6 and Table 7). It was observed that the performance of EEPVAH and TWC genotypes were inferior in terms of cob characteristics during season-2 than in season-1. Local check—DMH849 produced the longest (18.1 cm in season-1 and 17.1 cm in season-2) and bigger cob (5.1 cm in season-1 and 4.7 cm in season-2) during both seasons in the EEPVAH trial (Table 6). In the same trial, EEPVAH-67 recorded highest number of kernels per row (41 in season-1, and 39 in season-2) in both seasons. The genotype EEPVAH-41 had the highest number of kernel—rows per cob during season-1 while in season-2, EEPVAH-53 recorded the highest NOKRC (Table 6). The cob length (18.5 cm in season-1 and 17.1 cm in season-2) was recorded highest in A1803-37 in both seasons in the TWC trial. The local check produced highest NOKRC (17), and NOKPR (40), and cob length and diameter were statistically similar to top genotypes during season-1, however, the cob characteristics were inferior during season-2 (Table 7).

### 3.4. Stalk Lodging

Stalk lodging was statistically non-significant in both the trials during season-1, and the genotypes recorded up to 6% and 9% lodging in TWC and EEPVAH trials, respectively (Figure 3). Only a few genotypes (four) recorded lodging in the TWC trial in season-1 whereas lodging recorded was within 2 to 9% in 13 genotypes of the EEPVAH trial (Figure 3). The local check (DMH849) recorded comparatively low lodging; 2.8% in TWC season-1, and 6.6% in EEPVAH season-2 (Figure 3). In contrast, lodging was significantly higher during season-2 and statistically significant among the maize genotypes in both the trials. Three-way cross hybrid maize genotype A1803-37 produced 16% lodging (2 to 16% in overall), while the highest lodging recorded in EEPVAH genotypes was 20% (EEPVAH-55) (Figure 3).

### 3.5. Kernel Zinc and Total Carotenoid Content

Total carotenoid in EEPVAH genotypes ranged between 1.8 and 3.6 mg 100 g^−1^ while TWC genotypes had between 14.2 and 24.9 mg kg^−1^ (ppm) zinc in its kernels. Most of the hybrids in both the trials reported higher kernel—zinc and total carotenoid compared to the local check -DMH849 (Figure 4). The total carotenoid was highest in EEPVAH-46 (3.59 mg 100 g^−1^) followed by EEPVAH-55 (3.4 mg 100 g^−1^) and EEPVAH-51 (3.3 mg 100 g^−1^), in contrast, total carotenoid was ≤2 mg 100 g^−1^ in local check, EEPVAH-12 and Check (RE) genotypes (Figure 4). Three-way cross hybrid maize genotype A1831-8 reported the highest kernel zinc content (24.86 ppm) while DMH849 (local check), A1803-37, A1831-9, and A1830-14 had kernel—zinc content ≤ 16.5 ppm. It was observed that EEPVAH and TWC biofortified maize hybrids had 9.5 to 79.5%, and 2.5 to 52.3% higher total carotenoid and kernel—zinc content, respectively, compared to the local check (Figure 4).

### 3.6. Grain Yield

The tested EEPVAH maize genotypes were significantly different in terms of grain yield, producing 3.3 to 8.1 t ha^−1^ in season-1 and 1.4 to 4.7 t ha^−1^ during season-2. On the other hand, the yield ranges produced by TWC genotypes were 3.7 to 7.1 and 1.8 to 3.7 t ha^−1^, respectively, for season-1 and season-2 (Figure 5). The local check (DMH849—6.8 t ha^−1^ in TWC, and 8.1 t ha^−1^ in EEPVAH trial) out yielded all the introduced maize genotypes during season-1 (except TWC genotype A1831-3), however, the performance during season-2 was greatly reduced; TWC genotype by 52.7, and EEPVAH by 44.3% (Figure 5). The majority of introduced maize genotypes recorded a higher yield advantage over the local check in season-2; yield advantage over the local check was 1.6 to 54.5% in TWC genotypes while EEPVAH genotypes recorded even higher—4.8 to 97.9% (Figure 6). According to the AMMI analysis, TWC genotypes A1803-42, A1847-10, and A1803-13, and EEPVAH genotypes EEPVAH-8, EEPVAH-46, and EEPVAH-67, were found consistent in performance (in terms of grain yield) with higher average grain yield over the seasons and were placed nearest to the origin in the graph (Figure 7).

## 4. Discussion

### 4.1. Phenological Traits

The synchrony of tasseling and silking determines the outcomes of hybridization in maize. There is a higher possibility of seed setting when anthesis-silking days are closer as it increases effective pollination and eventually the grain yield [31]. Thus, narrowing the tasseling silking gap is of prime importance in hybrid development programs. As observed in the present study, Dhakal et al. [19] and Kandel and Shrestha [32] reported a non-significant effect on the anthesis-silking interval in the evaluated hybrid genotypes. It is observed that silking and tasseling takes double or more time in the winter season compared to summer or spring sowing [19,33], however, the tasseling and silking took a comparatively shorter duration in our experiment during season-2 (Table 2 and Table 3). In general, October sowing is practiced for winter cropping, and we tested August sowing in our experiment (August 31). The daily average temperature remained within 24 to 30 °C for up to mid-October after sowing (Figure 1), by the time tasseling was initiated in several genotypes in our experiments, thus shortened the tasseling and silking compared to February sowing, and other winter sowings (October/November).

Right from germination to different growth stages and eventually to the physiological maturity, temperature plays an important role in maize crop. Maize requires around a 20 °C to 30 °C temperature for the rapid emergence of seedlings, and 28 °C is considered optimum for both tasseling and silking in maize [34,35,36]. The emergence and overall growth of the plant is faster when an optimum temperature is provided, and it slows down when the threshold is not met [37]. In addition, the short duration nature of the variety obtains earlier fulfillment of growing degree days required for tasseling and silking [37]. Regarding the tasseling and silking in hybrid maize, the mean value ranged between 66 and 108 days, and between 70 and 119 days, respectively, under Terai and Inner-Terai regions during the winter season in Nepal [32,33,38,39]. Corroborating our results, anthesis and silking days in provitamin A tropical maize genotypes were reported at 55–65 days, and 56–69 days, respectively, from Africa [40,41].

### 4.2. Growth, Yield Attributing, and Cob Characteristics

The overall growth, yield attributing traits, and cob characteristics of maize genotypes were reduced during season-2 comparative to season-1. The physiological stress attributed to fall armyworm damage during the vegetative stage, and drought following tasseling had great effect in our experiment in season-2. Fall armyworm larvae, right from the first instar, is most damaging, and feed on foliage of maize plants. Larger larvae feed extensively and cause heavy defoliation, and at a higher damage level, only the ribs and stalk remain in the plant [42]. Thus, its infestation reduces the photosynthetic area, elevates physiological stress, and affects normal growth and development of the plants. From a study conducted to assess the effect of the fall armyworm at an early vegetative stage, it was reported that the late whorl stage was most sensitive, and mean larvae 0.2 to 0.8 per plant at this stage could result in a 5 to 20% yield reduction in maize [43].

In maize, the number of kernel—rows in the cob is determined at the V5 to V8 growth stage, which is controlled mostly by genetic factors, and to a lesser degree, by the growing environment [44,45]. Thus, the number of kernel—rows per cob (NOKRC) was found significantly different in EEPVAH genotypes in both the seasons. The non-significant effect during season-2, in TWC genotypes (Table 7), might be due to the influence of drought and physiological stress caused by fall armyworm infestation. On the other hand, the number of kernels per row (NOKPR), heavily influenced by the growing environment and crop management practices, is determined between the V12-V15 growth stages [46]. For this reason, we observed a reduction in the mean number of kernels per row (Table 6 and Table 7) during season-2 due to drought (Figure 1) and fall armyworm damage in our experiment (Figure 2). The observed reduction in cob size (cob length and diameter) during season-2 (Table 6 and Table 7), in both EEPVAH and TWC trials, was due to the corresponding reductions in NOKRC and NOKPR as the cob size of maize is influenced by these traits [45]. Prolonged drought (Figure 1) immediate after the silking stage during season-2 might have resulted in comparatively smaller and lighter grains in the cob (reported as hundred kernel weight in Table 4 and Table 5) due to less dry matter deposition in the kernels [44]. It was reported that long duration of drought stress following silking affects normal cob development and reduces kernel size and weight in maize [47,48]. The continued drought stress across several growth stages in maize attributes to a complete crop failure [48]. Thus, we observed a reduction in performance of the genotypes in terms of yield attributing traits (Table 4 and Table 5), cob characteristics (Table 6 and Table 7), and grain yield (Figure 5) during season-2 due to prolonged drought following silking to the crop maturity stages (Figure 1).

### 4.3. Stalk Lodging

The breaking of the stalk below the ear is recorded as stalk lodging in maize [26]. Stalk lodging, in addition to yield loss up to 5 to 25%, increases harvest loss, harvest time and decreases the quality of grains [49,50]. Genotypes with a short stature suffered less lodging as in the local check and other maize genotypes during season-1 (Table 4 and Table 5), however, season-2 lodging was exacerbated by the combined stress caused by fall armyworm infestation and drought. The continuous infestation of the fall armyworm from the early stage resulted in weaker (small stalk diameter) plants (Figure 2), and drought stress following tasseling resulted in higher lodging in season-2. Continuous infestation of fall armyworm feeding on vascular tissues of plant might have increased the physiological stress which favored the stalk lodging in plants [49]. Additionally, it is reported that drought stress during the grain filling period increases the potential for stalk lodging in maize [44,51]. In our experiment, the effect of fall armyworm infestation and drought stress in season-2 cropping is observed clearly in growth and yield attributing traits (Table 4 and Table 5), cob characteristics (Table 6 and Table 7), stalk lodging (Figure 3), and grain yield (Figure 5).

### 4.4. Kernel Zinc and Total Carotenoid

Discriminating the zinc content of tested genotypes in our experiment, it is evident that the local check had low zinc content in its kernel (16.3 mg kg^−1^), while some introduced genotypes were highly enriched (more than 50%). It can be inferred that about 500 g grain (not including cooking or other processing losses) of high zinc maize line (A1831- 8, 24.86 mg kg^−1^) could supply the daily recommended dietary intake for adults (up to 12 mg day^−1^). On the other hand, provitamin A enriched maize lines (total carotenoid—3.59 mg 100 g^−1^) seem promising to supply the daily dietary requirement of vitamin A; the adult human body requires 900 and 700 microgram (μg) retinol activity equivalents (RAE) per day, respectively, for men and women [8]. Thus, genotypes with a significantly higher amount of total carotenoid and zinc content were found in EEPVAH and TWC trials of maize compared to the local check which can be used as important genetic resources in future breeding programs and recommend for cultivation after further evaluation.

The yellow or orange endosperm of provitamin A contains carotenoids, which is the precursor of vitamin A [52]. The pollen gene from a non provitamin A parent affects development of fruit or seeds, and hence, kernel micronutrient content, which is known as the xenia effect [53]. The total carotenoid content, overall, in our experiment was lower (Figure 4) in comparison to the breeding target; 15 µg g^−1^ of β—carotene standard [54]. This might be attributed to the xenia effect of adjacent white kernel hybrids from the three-way cross hybrid maize trial which contained mixture of white and yellow kernel hybrids. Similarly, it has been demonstrated that drought also influences provitamin A and β—carotene content in maize. Ortiz-Covarrubias et al. [55] reported lower provitamin A (12.9 µg g^−1^) and β—carotene content (16.5% lower) under drought than that in optimum growing contitions (14.1 µg g^−1^). In another study, the β—carotene was reported within the range between 1.67 and 3.39 µg g^−1^ under drought conditions [52].

The maximum zinc content of the genotype in our experiment was 24.86 mg kg^−1^ (Figure 4), which is lower (32.81%) than the breeding target for zinc in maize—37 mg kg^−1^ on the dry matter basis [13]. As soil zinc availability greatly influences kernel-zinc content in maize, the lower zinc content in our experiment might be associated with low soil zinc levels. Researchers have reported that breeding efforts for increasing kernel zinc concentration is highly dependent on agronomic interventions for most of the developing countries report deficient zinc levels in soil [13]. As a proof, several papers have reported widespread zinc deficiency in different agro-ecology of Nepalese soils, including similar to our study area [56,57,58,59,60].

### 4.5. Grain Yield

Grain yield is a key parameter for identifying and recommending genotypes for cultivation at the farmer level. The performance of the local check, in season-1, was superior to all the introduced maize genotypes, while in season-2, a drastic reduction was observed (Figure 5). The reduction in overall performance of the genotypes could be associated with physiological stress (caused by fall armyworm damage) during the vegetative stage, and drought following the tasseling stage. No supplementary irrigation was provided and there was no rainfall recorded (Figure 1) following tasseling in season-2. After the entry (9 May 2019) in Nepal [25], the fall armyworm has been widespread and its infestation was observed continuous in season-2 (August sowing), higher before the tasseling stage, in our experiment (Figure 2). The intensity was higher as there was no maize cultivation around the study location and the chemical control was ineffective. Thus, overall growth and yield attributing (Table 4 and Table 5), cob characteristics (Table 6 and Table 7), and grain yield of maize (Figure 5) were significantly reduced in both the trials during season-2. Fall armyworm larvae at a mean density of 0.2 to 0.8 per plant could reduce grain yield by 5 to 20%, if it occurred at the late whorl stage [43]. The reduction in yield was higher in the TWC trial than in EEPVAH, about a 10% reduction (Figure 5). This might be due to the genotypes in the EEPVAH trial being stress tolerant as stated by IITA, Nigeria.

In a recent study conducted by the National Maize Research Program, Rampur, Nepal, the highest mean grain yield obtained from provitamin A biofortified maize, combining the spring and winter season, was 8.2 t ha^−1^ (crop geometry—0.75 m × 0.20 m and fertilizer dose of 180:60:40 NP_2_O_5_K_2_O ha^−1^). It was also reported that the provitamin A hybrids yielded 3.5 to 8.5 t ha^−1^ in spring, while the yield range was higher in winter—3 to 10.6 t ha^−1^ [33]. In the same location, the mean grain yields of the top four three-way cross white and yellow kernel hybrid (non-biofortified) maize were 9.5 and 8.8 t ha^−1^ in the winter season (crop geometry—0.60 m × 0.25 m and fertilizer dose of 180:60:40 NP_2_O_5_K_2_O ha^−1^) [61]. Similarly, the mean yield of the top performing three-way cross (non- biofortified) hybrid reported from Surkhet (current study location) was 8.4 t ha^−1^ in winter and 9.8 t ha^−1^ in summer [19]. The provitamin A tropical maize inbreed lines were reported to yield up to 6.2 t ha^−1^ on average (range—3.9 to 7.9 t ha^−1^) in Africa [40]. From the recent study, it was also disseminated that three-way cross provitamin A hybrids yielded between 1.1 and 5.1 t ha^−1^ in Nigeria [41]. The grain yield reported in season-1, in our study, corroborates with the findings of the previous studies and even indicates the scope of increasing it by maintaining plant population and fertilization as per recommendation.

Drought and fall armyworm infestation significantly reduced the grain yield in season-2 in our experiment (Figure 5). It was reported that the fall armyworm can reduce maize grain yield by 5–40% and even cause heavy losses at the higher infestation level [62,63,64,65]. In Africa, annual maize production loss due to fall armyworm damage was estimated at 25 to 53%, which was around 2.48 to 6.19 billion USD [66]. Similarly, researchers reported that prolonged drought stress following silking affects normal cob development, and reduces kernel size and weight [47,48], and moderate to extreme drought could reduce grain yield in maize by 64 to 74% [67].

## 5. Conclusions

The provitamin A genotypes EEPVAH-46 and EEPVAH-51 (total carotenoid 3.6 and 3.3 mg 100 g^−1^), and zinc biofortified genotypes A1847-10 and A1803-42 (20.4 and 22.4 mg kg^−1^ zinc) were identified as superior genotypes based on their yield consistency over the seasons and higher provitamin A and zinc content compared to the local check. The plant population in the present experiment was almost half (40–50 cm intra—row), in comparison to the government recommendation (25 cm for hybrids), and we applied 120:60:40 NP_2_O_5_K_2_O ha^−1^ instead of 180:60:40 NP_2_O_5_K_2_O ha^−1^ (for hybrids), thus there is scope to increase the grain yield by maintaining plant population and fertilizer doses. The new sowing time (August) was tested for winter harvesting of maize, which enables farmers to harvest green cobs during December–January. Additionally, micronutrient rich maize is highly desired in the poultry industry to prepare feeds rich in vitamins and minerals. Thus, the identified provitamin A and high zinc maize genotypes could equally be effective in reducing hidden—hunger, enhancing feed nutrient value for poultry and livestock sectors, and also has commercial value in a green cob business.

## Figures and Tables

**Figure 1 plants-11-02898-f001:**
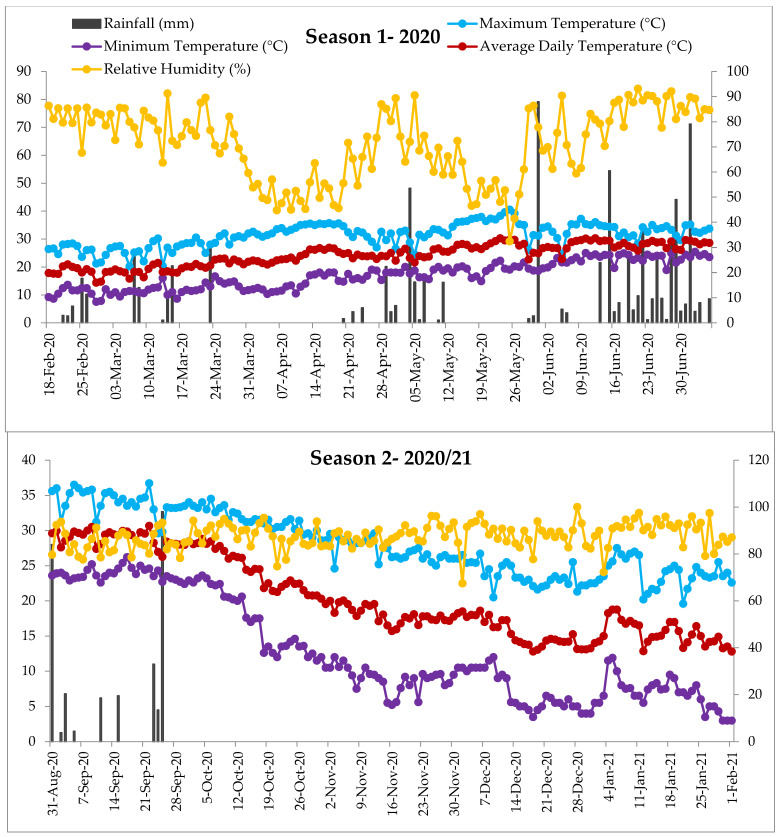
Agro—meteorological parameters recorded during season-1 (2020) and season-2 (2020/21). Daily rainfall (bar, black), minimum (purple line with markers), average (dark red line with markers), and maximum temperature (light-blue line with markers) are represented in the primary vertical axis, whereas relative humidity (Orange line with markers) is represented through the secondary vertical axis. The date is presented in day-month-year format where tail digit 20 (as in 1 September 2020) and 21 (as in 26 January 2021) indicate years 2020 and 2021, respectively. Meteorological data were obtained from the nearest meteorological station (Mehelkuna site) of the office of hydrology and meteorology.

**Figure 2 plants-11-02898-f002:**
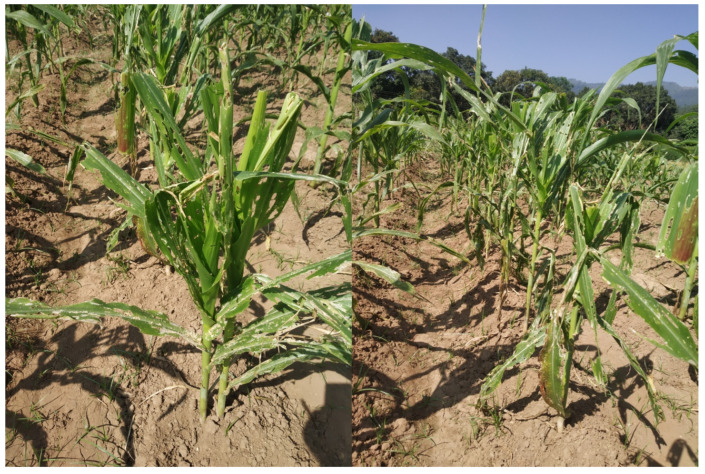
Fall armyworm damage observed in maize trials during season-2. The photo was taken between 41 and 47 days after sowing. The maize plants were continuously under infestation right from the seedling emergence until fully exertion of the tassel, pesticide application was in-effective, there was no maize cultivation near the study location. Plants were under physiological stress, thus vegetative growth was inferior compared to the normal season (season-1).

**Figure 3 plants-11-02898-f003:**
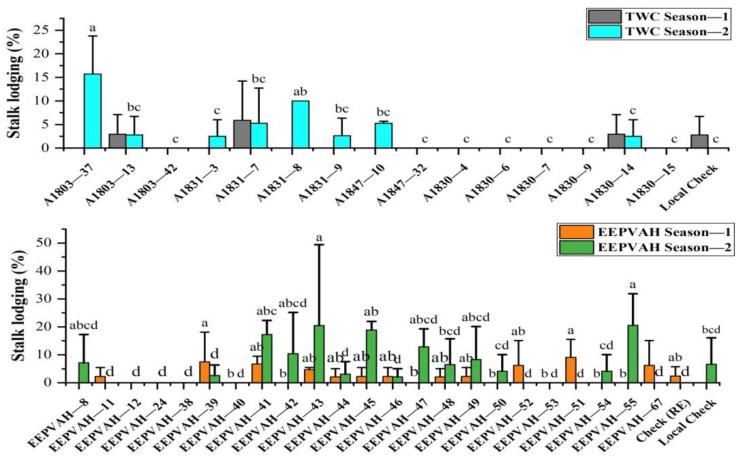
Stalk lodging recorded from maize genotypes of EEPVAH and TWC trials conducted in two contrasting seasons. Similar statistical letters for a season indicates non-significant effect, genotypes having only statistical letters and without a visible bar indicates no report of stalk lodging (zero lodging). Error bar reports only the positive values.

**Figure 4 plants-11-02898-f004:**
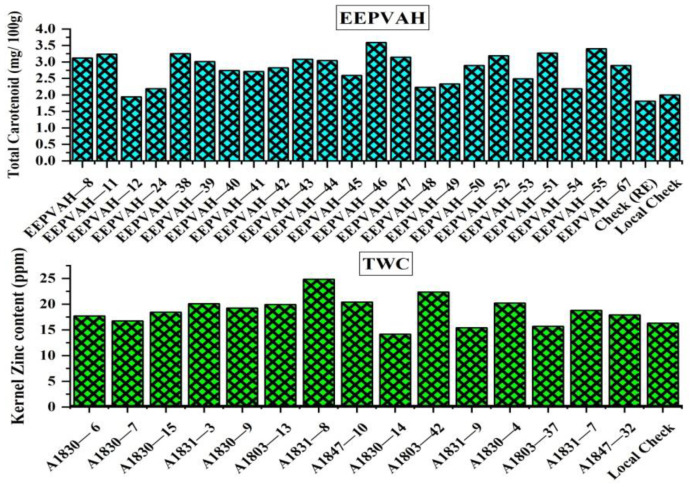
Total carotenoid and kernel zinc content of biofortified maize hybrids from the EEPVAH and TWC trials conducted in season-2. Composite samples were taken from each genotype in both the trials and submitted to the laboratory within 30 days of harvesting of the crop.

**Figure 5 plants-11-02898-f005:**
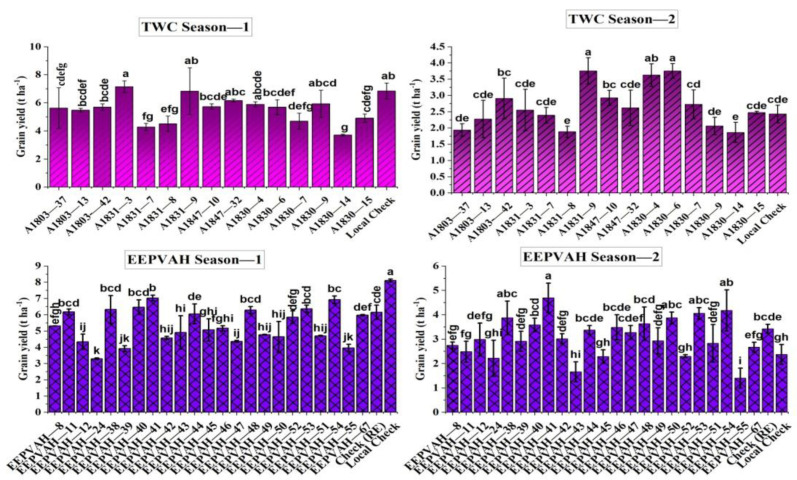
Grain yield recorded in the two different maize trials conducted in the contrasting seasons in the river basin area of Karnali Province, Nepal. TWC indicates three-way cross zinc biofortified hybrid maize trial, EEPVAH indicates regional extra-early multi—stress tolerant provitamin A biofortified hybrid maize trial. DMH849 was the local check in both the trials. Same statistical letters within the trial and season indicate a non-significant effect.

**Figure 6 plants-11-02898-f006:**
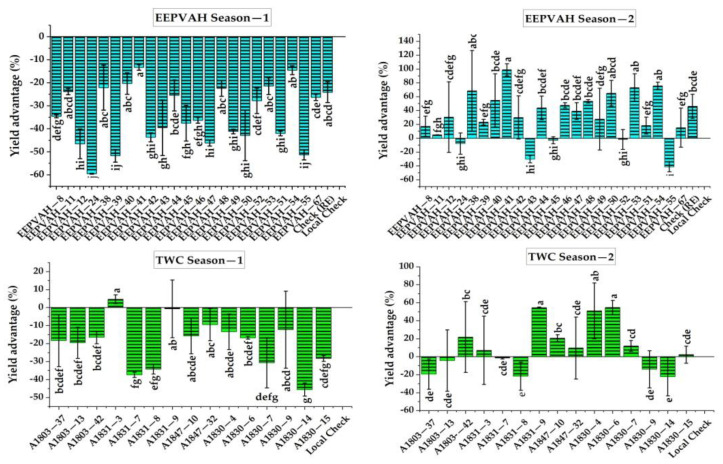
Yield advantage of TWC and EEPVAH genotypes over the local check recorded from two contrasting seasons. The downward bar indicates a negative yield advantage over the grain yield of the local check (DMH849). Similar statistical letters among the genotypes within the season indicates non-significant effect.

**Figure 7 plants-11-02898-f007:**
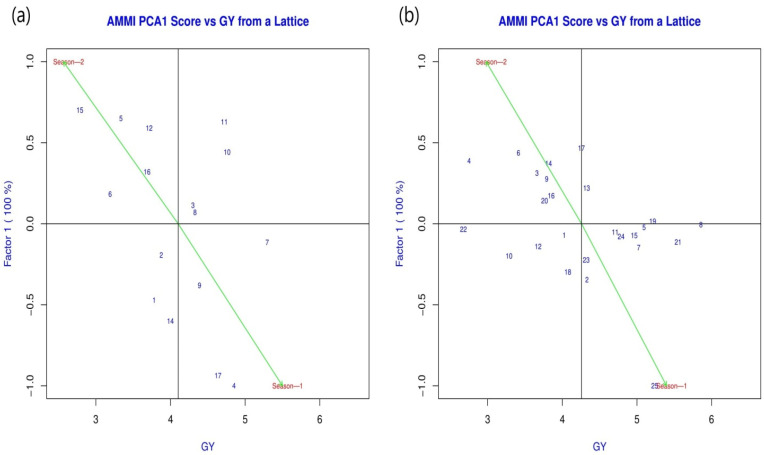
AMMI analysis of the grain yield trait of the maize genotypes from the two maize trials conducted under the contrasting seasons; figure (**a**) represents AMMI analysis of three way cross zinc biofortified hybrid maize genotypes, and figure (**b**) represents AMMI analysis of the regional extra early provitamin A biofortified hybrid maize genotypes, GY—grain yield (t ha^−1^). The numerical value in the figure indicates respective entry numbers of the genotypes, the name of which can be retrieved from Table 1.

**Table 1 plants-11-02898-t001:** Details of the genetic materials used in extra early multi-stress tolerant provitamin A and three-way cross zinc biofortified hybrid maize trials.

EEPVAH Genotypes					
EN	Designation	EN	Designation	EN	Designation
1	EEPVAH-8	10	EEPVAH-43	19	EEPVAH-53
2	EEPVAH-11	11	EEPVAH-44	20	EEPVAH-51
3	EEPVAH-12	12	EEPVAH-45	21	EEPVAH-54
4	EEPVAH-24	13	EEPVAH-46	22	EEPVAH-55
5	EEPVAH-38	14	EEPVAH-47	23	EEPVAH-67
6	EEPVAH-39	15	EEPVAH-48	24	Check (RE)
7	EEPVAH-40	16	EEPVAH-49	25	DMH849
8	EEPVAH-41	17	EEPVAH-50		
9	EEPVAH-42	18	EEPVAH-52		
**TWC Genotypes**					
**EN**	**Designation**	**Source**	**EN**	**Designation**	**Source**
1	A1803-37	18C30096B	9	A1847-32	18C30116B
2	A1803-13	18C30098B	10	A1830-4	18C30122B
3	A1803-42	18C30100B	11	A1830-6	18C30124B
4	A1831-3	18C30102B	12	A1830-7	18C30126B
5	A1831-7	18C30104B	14	A1830-9	18C30130B
6	A1831-8	18C30106B	15	A1830-14	18C30132B
7	A1831-9	18C30108B	16	A1830-15	18C30134B
8	A1847-10	18C30110B	17	DMH849	Local agrovet

Remarks: EN: entry number, EEPVAH: regional extra early multi-stress tolerant provitamin A biofortified maize lines, check (RE): pedigree—TZEE-Y Pop STR C5 × TZEEI 58, extra-early in maturity, released in Ghana, Nigeria and Mali, potential yield and provitamin A content are 5 t ha^−1^ and 11.4 μg g^−1^, respectively, TWC: three-way cross zinc biofortified hybrid maize lines, DMH849 in both the trial was used as local check which indicates widely grown hybrid maize around the experimental area.

**Table 2 plants-11-02898-t002:** Phenological traits of extra early provitamin A biofortified maize genotypes recorded from two contrasting seasons.

Genotypes	Days to Tasseling		Days to Silking		Anthesis-Silking Interval	
	Season-1	Season-2	Season-1	Season-2	Season-1	Season-2
EEPVAH-8	74 efg	53 bc	76 efg	57 bc	2	4
EEPVAH-11	72 hi	49 f	74 hi	53 h	1	4
EEPVAH-12	72 i	50 ef	74 hi	54 gh	2	4
EEPVAH-24	74 def	55 ab	76 def	59 ab	2	4
EEPVAH-38	72 i	51 cdef	74 hi	56 cdef	2	4
EEPVAH-39	74 efg	51 def	76 efg	57 cd	2	5
EEPVAH-40	73 fgh	51 cdef	75 fg	55 defg	2	4
EEPVAH-41	74 efg	52 cde	76 efg	54 gh	2	2
EEPVAH-42	74 efg	52 cd	76 efg	57 cd	2	4
EEPVAH-43	76 bc	55 ab	78 bc	59 ab	2	4
EEPVAH-44	73 fgh	52 cd	75 gh	56 cdef	1	3
EEPVAH-45	76 b	51 def	78 b	55 efgh	2	4
EEPVAH-46	75 cde	51 def	77 cde	54 gh	2	3
EEPVAH-47	73 ghi	50 ef	75 gh	55 efgh	2	5
EEPVAH-48	74 efg	52 cd	76 efg	55 efgh	2	2
EEPVAH-49	78 a	51 def	80 a	55 efgh	2	4
EEPVAH-50	75 bcd	52 cde	77 bcd	56 cde	2	4
EEPVAH-52	73 fgh	52 cde	75 fg	56 cde	2	4
EEPVAH-53	75 bcd	51 cdef	77 bcd	56 cdef	2	4
EEPVAH-51	73 fgh	50 def	75 fg	55 efgh	2	4
EEPVAH-54	75 cde	52 cde	77 cde	55 defg	2	3
EEPVAH-55	75 bcd	51 def	77 bcd	55 defg	2	4
EEPVAH-67	74 def	52 cd	76 def	57 cd	2	4
Check (RE)	70 j	51 cdef	73 i	54 fgh	2	3
Local Check	73 fgh	56 a	75 fg	59 a	2	3
Mean	74	52	76	56	2	4
Std	0.7	1	0.7	0.9	0.2	0.7
*p*-value	<0.01	<0.01	<0.01	<0.01	0.2	0.1
StdMSE	0.7	0.9	0.7	0.9	0.2	0.7
CV (%)	0.9	1.9	0.9	1.6	10.7	18.8

Remarks: StdMSE—standard mean sum of error, Std—standard deviation, LSD—least significant difference, CV—coefficient of variation, *p*-value < 0.01 indicates highly significance, <0.05 significant effect, >0.05—non-significant effect, similar statistical letters within a parameter (column) indicates non-significant difference among the genotypes.

**Table 3 plants-11-02898-t003:** Phenological traits of three-way cross zinc biofortified maize genotypes recorded from two contrasting seasons.

Genotypes	Days to Tasseling		Days to Silking		Anthesis-Silking Interval	
	Season-1	Season-2	Season-1	Season-2	Season-1	Season-2
A1803-37	82 bcde	56 cdef	84 bcd	59 def	2	3
A1803-13	81 cde	59 b	83 bcd	63 bc	2	4
A1803-42	82 bcde	57 bcd	84 bc	61 bcde	2	4
A1831-3	84 abc	59 b	86 ab	63 bcd	2	3
A1831-7	82 bcde	56 cde	84 bc	60 bcdef	2	4
A1831-8	84 abcd	58 bc	86 ab	64 ab	2	5
A1831-9	79 e	55 def	81 cd	59 ef	2	3
A1847-10	79 ef	53 f	81 de	57 f	2	4
A1847-32	79 e	54 ef	82 cd	58 ef	2	4
A1830-4	82 bcde	57 bcde	84 bcd	61 bcdef	2	4
A1830-6	81 de	56 cde	83 cd	60 cdef	2	3
A1830-7	81 cde	55 def	84 bcd	59 ef	2	3
A1830-9	82 bcde	57 bcd	84 bcd	60 bcdef	2	3
A1830-14	86 a	62 a	88 a	67 a	2	5
A1830-15	85 ab	56 cdef	88 a	59 def	3	3
Local check	76 f	55 def	78 e	59 ef	2	4
Mean	82	57	84	61	2	4
Std	1.5	1.4	1.5	1.6	0.3	0.7
*p*-value	< 0.01	< 0.01	< 0.01	< 0.01	0.3	0.2
StdMSE	1.5	1.4	1.5	1.7	0.4	0.7
CV (%)	1.8	2.4	1.8	2.7	17.3	17.4

Remarks: StdMSE—standard mean sum of error, Std—standard deviation, LSD—least significant difference, CV—coefficient of variation, *p*-value < 0.01 indicates highly significance, <0.05 significant effect, >0.05—non-significant effect, similar statistical letters within a parameter (column) indicates non-significant difference among the genotypes.

**Table 4 plants-11-02898-t004:** Growth and yield attributing traits recorded in provitamin A biofortified maize genotypes from the two contrasting seasons.

Genotypes	Plant Height (cm)		Ear Height (cm)		Plants Harvested		Ears Harvested		Shelling (%)		Harvest Index		Hundred Kernel Weight (gm)	
	Season-1	Season-2	Season-1	Season-2	Season-1	Season-2	Season-1	Season-2	Season-1	Season-2	Season-1	Season-2	Season-1	Season-2
EEPVAH-8	196 fg	186.6 hi	99 defghi	88.4 cdefg	21	15 d	36 abcde	29 defg	84.9 cdefgh	82.4 bcdefghij	0.6 a	0.7 a	29.8 bcd	28 bcd
EEPVAH-11	198.5 fg	178.2 i	105.2 bcdef	77.2 ghi	22	16 cd	38 abcd	22 gh	85.7 bcde	83.2 abcdefgh	0.5 cd	0.6 bc	31.5 b	31.3 ab
EEPVAH-12	211.4 de	204.8 cdefg	102.3 cdefg	90.7 bcdef	21	22 a	26 hi	29 defg	83.9 fgh	81.3 efghijk	0.4 j	0.4 efgh	31.1 bc	27 cde
EEPVAH-24	197.8 fg	196.7 fgh	95.4 efghij	88.9 cdefg	18	15 d	23 i	22 gh	81.9 ij	82.1 cdefghij	0.5 defgh	0.6 bcd	28.1 def	25.3 defg
EEPVAH-38	200.5 efg	198 fgh	104.5 bcdef	93 bcdef	22	22 a	39 abc	35 bcd	85.9 bcde	83.1 bcdefgh	0.5 cd	0.5 cde	29.8 bcd	26.3 def
EEPVAH-39	220.3 cd	213.6 bcdef	117 ab	98.3 bc	21	21 ab	26 hi	26 fgh	85.2 cdefg	83.8 abcdefg	0.3 k	0.4 ijk	28.7 def	27.4 cd
EEPVAH-40	220.8 cd	219 bc	101.8 cdefg	90.1 bcdef	22	21 ab	41 a	39 ab	86.2 bcd	85.5 ab	0.5 fgh	0.4 ghij	31.1 bc	27.7 bcd
EEPVAH-41	213.4 d	211 bcdefg	83.6 j	84.3 efgh	22	22 a	39 abc	39 ab	87 ab	86.5 a	0.5 cd	0.5 bcd	27.9 def	24.6 defg
EEPVAH-42	214 cd	201.2 defgh	106.7 bcde	84.3 efgh	23	22 a	30 gh	26 fgh	86.3 bc	84.9 abcd	0.4 j	0.4 hijk	28.6 def	27.6 bcd
EEPVAH-43	242 a	221.8 bc	107.8 bcde	98 bc	21	22 a	40 ab	33 bcdef	81.3 j	76.5 l	0.4 j	0.3 k	25.3 hi	19.6 i
EEPVAH-44	199.7 efg	200.5 efgh	102.2 cdefg	85.2 defgh	23	19 bc	35 bcdefg	28 efg	84.4 efgh	84 abcdefg	0.5 cde	0.6 b	35.6 a	33.8 a
EEPVAH-45	217.6 cd	216.2 bcde	110.7 bcd	85 d efgh	22	21 ab	37 abcd	29 cdefg	84.9 cdefgh	83.6 abcdefgh	0.4 j	0.4 ghij	25.8 ghi	24.2 defgh
EEPVAH-46	217.2 cd	211.9 bcdefg	111.6 bcd	98.2 bc	22	22 a	36 abcde	33 bcde	84.3 efgh	82.8 bcdefghi	0.4 ij	0.4 ijk	26.8 fgh	27.3 cd
EEPVAH-47	212 de	217.9 bcd	100.4 defghi	96 bcde	23	22 a	30 fgh	31 cdef	84.5 defgh	85.3 abc	0.4 ij	0.4 jk	29.1 cde	27.8 bcd
EEPVAH-48	211.8 de	200.6 efgh	87.7 hij	87.5 cdefg	24	22 a	39 abc	39 ab	81.4 j	80.5 hijk	0.5 cdefg	0.5 defg	25.3 hi	21.5 ghi
EEPVAH-49	226.4 bc	223.1 b	108.2 bcde	101.6 b	21	20 ab	34 cdefg	28 efg	83.8 gh	82.8 bcdefghi	0.4 ij	0.6 b	24.9 hi	22.7 fghi
EEPVAH-50	195.4 g	197.1 fgh	107.4 bcde	97 bcd	23	22 a	35 bcdefg	27 efgh	84.3 efgh	84.1 abcdef	0.5 ghi	0.4 hijk	24.8 hi	30.1 abc
EEPVAH-52	215.1 cd	195.8 gh	107.2 bcde	84.9 defgh	24	21 ab	41 a	28 defg	85.5 bcdef	80.8 ghijk	0.5 efgh	0.4 fghi	23.7 i	20.5 hi
EEPVAH-53	198 fg	185.1 hi	86.7 ij	83.1 fgh	23	22 a	39 abc	36 abc	86.4 bc	84.4 abcde	0.5 bc	0.5 cde	27.6 efg	25 defg
EEPVAH-51	198.1 fg	200.7 defgh	91.6 fghij	70.6 i	22	20 ab	33 defg	28 efg	85.2 cdefg	81.7 defghij	0.5 cdef	0.5 cdef	28.8 def	25.1 defg
EEPVAH-54	213.3 d	201.4 defgh	100.9 defgh	90.4 bcdef	22	22 a	40 ab	43 a	88.5 a	84.9 abcd	0.5 bc	0.6 b	28.4 def	23.3 efghi
EEPVAH-55	220.1 cd	204.9 cdefg	103.5 bcdefg	83.4 fgh	18	22 ab	31 efgh	29 cdefg	83.4 hi	79.2 jkl	0.5 hi	0.5 efgh	27.4 efg	22.5 fghi
EEPVAH-67	236.9 ab	244 a	127.1 a	116.5 a	24	21 ab	40 ab	26 fgh	84.4 efgh	79.7 ijkl	0.4 ij	0.4 hijk	29.4 cde	27 cde
Check (RE)	220.2 cd	212.6 bcdefg	115.6 abc	86.5 cdefg	21	22 a	32 defg	32 bcdef	84.5 defgh	81.1 fghijk	0.5 fgh	0.5 defg	33.9 a	30.7 abc
Local Check	208.5 def	186.6 hi	89.8 ghij	74.1 hi	23	15 d	36 abcdef	20 h	83.9 fgh	78.4 kl	0.6 ab	0.5 bcd	35.8 a	32.4 a
Mean	212.2	205.2	102.9	89.3	22	20	35	30	84.7	82.5	0.5	0.5	28.8	26.4
Std	5.9	8	6.5	5.7	1.5	1.7	2.7	3.3	0.8	1.6	0.02	0.04	1	1.8
*p*-value	<0.01	<0.01	<0.01	<0.01	0.2	0.01	<0.01	<0.01	<0.01	<0.01	<0.01	<0.01	<0.01	<0.01
StdMSE	5.9	8.1	6.5	5.8	1.5	1.8	2.7	3.4	0.8	1.6	0.02	0.04	0.9	1.8
CV (%)	2.8	3.9	6.3	6.4	6.7	8.6	7.6	11.1	0.9	1.9	4.1	7.7	3.4	6.9

Remarks: StdMSE—standard mean sum of error, Std—standard deviation, LSD—least significant difference, CV—coefficient of variation, *p*-value < 0.01 indicates highly significance, <0.05 significant effect, >0.05—non-significant effect, similar statistical letters within a parameter (column) indicates non-significant difference among the genotypes.

**Table 5 plants-11-02898-t005:** Growth and yield attributing traits recorded in zinc biofortified maize genotypes from the two contrasting seasons.

Genotypes	Plant Height (cm)		Ear Height (cm)		Plants Harvested		Ears Harvested		Shelling (%)		Harvest Index		Hundred Kernel Weight (gm)	
	Season-1	Season-2	Season-1	Season-2	Season-1	Season-2	Season-1	Season-2	Season-1	Season-2	Season-1	Season-2	Season-1	Season-2
A1803-37	219.2 cdef	188.7 def	114.3 cde	67.9 g	19 ab	12 d	33 abcde	16 e	82.7 cde	77.5 abcd	0.4 bc	0.6 a	36.5 a	30.7
A1803-13	254.1 ab	208.3 a	126.9 bc	92.3 ab	18 bc	18 ab	30 defgh	22 cd	84.3 abc	78 abc	0.3 de	0.4 e	30.9 f	27.7
A1803-42	261 a	203.1 abc	136.3 ab	81.6 bcdef	18 ab	19 a	31 cdefg	27 b	81.6 defg	74.5 de	0.3 e	0.5 cde	30.5 f	29.5
A1831-3	256.2 ab	180.9 efg	151 a	75.4 efg	19 ab	20 a	34 abcd	21 d	80.8 efgh	76.5 cd	0.3 de	0.5 cde	36.2 a	29.7
A1831-7	217.1 cdef	186.2 efg	119.3 bcde	85.3 abcde	18 bc	19 a	26 gh	28 b	83.7 abcd	77.5 abcd	0.3 de	0.5 bcd	31.4 ef	26.1
A1831-8	220.6 cdef	188.8 cdef	117.8 bcde	88.6 abcd	19 ab	20 a	30 cdefgh	27 b	82.7 cde	77 bcd	0.3 e	0.5 cde	31 f	22.8
A1831-9	228.8 cd	203.6 ab	121.5 bcd	95.7 a	19 ab	19 a	35 abc	32 a	85.6 ab	80.5 a	0.4 bcd	0.5 abcd	33.3 cd	29.1
A1847-10	220.6 cdef	184.5 efg	114.2 cde	85.5 abcde	18 ab	19 a	32 cdef	27 b	86.1 a	79.5 abc	0.5 b	0.5 abc	32.4 de	28
A1847-32	235.1 bc	189.5 bcdef	119.4 bcde	81.2 bcdef	20 ab	19 a	32 bcdef	22 cd	83.1 bcde	79.5 abc	0.4 bcde	0.5 cde	33 cd	28
A1830-4	223.4 cde	200.9 abcd	100.4 ef	85.6 abcde	18 ab	20 a	28 fgh	26 bc	79.8 fgh	79 abc	0.4 cde	0.6 ab	36.4 a	29.1
A1830-6	200 f	175.3 fg	101.8 def	79 cdefg	20 a	16 bc	32 bcdef	27 b	82.2 cdef	76.5 cd	0.4 cde	0.5 cde	36.5 a	30.1
A1830-7	218.8 cdef	194.4 abcde	117.9 bcde	89.3 abc	18 bc	18 ab	28 efgh	25 bcd	83.2 bcde	80 ab	0.3 e	0.5 cde	32.9 cd	29.8
A1830-9	226.7 cd	172.4 g	116.5 bcde	69.9 fg	19 ab	18 ab	37 ab	21 d	81.1 defgh	74.5 de	0.3 e	0.5 abc	31.2 ef	26.5
A1830-14	224.4 cde	180.5 efg	112.9 cde	75.6 defg	16 c	20 a	26 h	25 bcd	78.7 h	73 e	0.3 e	0.4 de	34.1 bc	26.1
A1830-15	212.7 def	190.2 bcde	110.3 cde	78.3 cdefg	19 ab	15 c	38 a	21 d	79 gh	77.5 abcd	0.3 e	0.5 bcd	30.9 f	29.2
Local check	202.5 ef	193.9 bcde	88.7 f	71.2 fg	18 bc	19 a	30 cdefgh	25 bcd	82.9 cde	78.5 abc	0.6 a	0.6 ab	34.7 b	30.6
Mean	226.3	190.1	116.8	81.4	19	18	32	25	82.3	77.5	0.4	0.5	33.2	28.3
Std	9.7	6.3	8.8	5.8	0.9	1.1	2.3	1.9	1.1	1.5	0.04	0.04	0.6	1.9
*p*-value	0.01	0.03	0.01	0.04	0.04	0.01	0.02	<0.01	0.02	0.05	<0.01	0.04	<0.01	0.3
StdMSE	9.8	6.3	8.8	5.8	0.9	1.1	2.3	1.9	1.2	1.4	0.04	0.1	0.6	2
CV (%)	4.3	3.3	7.5	7.1	4.7	6.3	7.3	7.6	1.4	1.9	10.7	8.5	1.8	6.9

Remarks: StdMSE—standard mean sum of error, Std—standard deviation, LSD—least significant difference, CV—coefficient of variation, *p*-value < 0.01 indicates highly significance, <0.05 significant effect, >0.05—non-significant effect, similar statistical letters within a parameter (column) indicates non-significant difference among the genotypes.

**Table 6 plants-11-02898-t006:** Cob characteristics recorded in provitamin A biofortified maize genotypes from the two contrasting seasons.

Genotypes	Cob Length (cm)		Cob Diameter (cm)		No. of Kernel- Rows per Cob		No. of Kernels per Row	
	Season-1	Season-2	Season-1	Season-2	Season-1	Season-2	Season-1	Season-2
EEPVAH-8	15.3 efghi	14.2 fgh	4.4 defghi	4.2 cde	13 h	12 efg	35 efg	34 defgh
EEPVAH-11	14.8 hij	14.7 efgh	4.5 cde	4.3 cd	14 fgh	12 efg	34 g	35 cde
EEPVAH-12	15.8 cde	14.8 defgh	4.5 cdef	4.3 cd	13 gh	13 def	37 cdefg	36 bcde
EEPVAH-24	15.5 defgh	14.5 efgh	4.3 ghijk	4 fgh	14 fgh	12 fg	38 bcde	36 bcde
EEPVAH-38	16 bcd	15.7 abcdef	4.4 cdefgh	4.4 bc	14 efgh	14 bcd	37 cdef	38 abc
EEPVAH-39	14.8 ij	14.5 efgh	4.3 defghij	4.3 cd	14 fgh	14 bcd	37 cdefg	34 defg
EEPVAH-40	15.1 fghij	14 gh	4.4 cdefg	4.2 cd	14 efgh	13 cde	36 cdefg	32 gh
EEPVAH-41	16.5 b	15.8 abcde	4.6 bc	4.3 cd	18 a	14 abcd	37 cdefg	37 abcd
EEPVAH-42	14.5 j	13.5 h	4.4 cdefg	4.2 cde	16 bc	13 def	36 defg	32 fgh
EEPVAH-43	15.1 fghij	13.8 h	4.1 klm	3.6 i	15 cdef	12 fg	35 fg	33 efgh
EEPVAH-44	15.7 cde	14.2 fgh	4.8 b	4.6 ab	13 gh	12 efg	35 defg	32 fgh
EEPVAH-45	15.4 defgh	14.5 efgh	4.2 ijkl	4.1 efgh	15 cdef	13 cde	39 abc	35 def
EEPVAH-46	14.7 ij	14.4 efgh	4.2 hijk	4.1 defg	15 cdef	13 cde	37 cdef	34 defgh
EEPVAH-47	14.9 ghij	14.9 cdefgh	4.3 fghijk	4.2 def	14 fgh	12 efg	37 cdef	35 def
EEPVAH-48	15.5 defg	14.9 cdefgh	4.4 defghi	4.2 def	16 b	15 abc	35 defg	34 defg
EEPVAH-49	15.9 bcd	16.4 abc	4.1 lm	3.8 h	15 bcde	14 abcd	39 abc	38 ab
EEPVAH-50	14.6 j	14.9 cdefgh	4.3 defghi	4.3 cd	16 b	15 ab	35 defg	34 defg
EEPVAH-52	15.3 efghi	14.5 efgh	4.1 jklm	3.9 h	16 bcd	14 bcd	37 cdefg	36 abcde
EEPVAH-53	15.4 defgh	14.3 efgh	4.4 defghi	4.3 cd	16 b	15 a	36 defg	34 efgh
EEPVAH-51	16.3 bc	16.3 abcd	4.2 ijkl	3.9 h	14 defg	13 def	35 defg	35 bcde
EEPVAH-54	15.7 cde	15.4 bcdefg	4.3 efghijk	3.9 gh	15 bcde	13 def	38 cdef	38 abc
EEPVAH-55	15.6 cdef	14.1 gh	4.1 m	3.4 i	13 h	11 g	38 bcd	31 h
EEPVAH-67	15 ghij	16.4 abc	4.5 cd	4.4 bcd	14 efgh	13 def	41 a	39 a
Check (RE)	17.7 a	16.5 ab	4.3 defghi	4.3 cd	14 efgh	13 cde	36 cdefg	34 defg
Local Check	18.1 a	17.1 a	5.1 a	4.7 a	16 bc	14 abcd	40 ab	38 ab
Mean	15.6	14.9	4.4	4.1	15	13	37	35
Std	0.3	0.7	0.09	0.1	0.6	0.6	1.3	1.4
*p*-value	<0.01	0.01	<0.01	<0.01	<0.01	<0.01	0.01	<0.01
StdMSE	0.3	0.7	0.1	0.1	0.6	0.6	1.3	1.4
CV (%)	1.9	4.9	2.1	2.4	4.3	4.8	3.6	4

Remarks: StdMSE—standard mean sum of error, Std—standard deviation, LSD—least significant difference, CV—coefficient of variation, *p*-value < 0.01 indicates highly significance, <0.05 significant effect, >0.05—non-significant effect, similar statistical letters within a parameter (column) indicates non-significant difference among the genotypes.

**Table 7 plants-11-02898-t007:** Cob characteristics recorded in zinc biofortified maize genotypes from the two contrasting seasons.

Genotypes	Cob Length (cm)		Cob Diameter (cm)		No. Of Kernel-Rows per Cob		No. of Kernels per Row	
	Season-1	Season-2	Season-1	Season-2	Season-1	Season-2	Season-1	Season-2
A1803-37	18.5 a	17.1 a	4.7 def	4.6	14 e	13	37 bcde	37 ab
A1803-13	16.6 bcde	14.9 bcd	4.7 bcde	4.2	16 bc	13	40 abc	33 cde
A1803-42	15.9 cde	14.1 cde	4.8 bcd	4.8	14 e	14	39 abc	30 ef
A1831-3	17.6 abc	15.4 abc	4.9 b	4.7	15 cd	14	37 abcde	34 bcd
A1831-7	15.3 defg	13.2 de	4.6 def	4.6	15 d	14	36 cde	33 def
A1831-8	15.3 defg	13.3 de	4.6 ef	4.1	15 cd	13	37 abcde	33 cde
A1831-9	16.6 bcde	15.3 abc	4.6 ef	4.4	14 e	13	39 abc	36 abc
A1847-10	16.9 abcd	15.1 bcd	4.3 g	4.3	14 efg	13	35 def	34 bcd
A1847-32	16.2 cde	15.1 bcd	4.8 bcd	4	15 cd	12	38 abcd	34 bcd
A1830-4	17.1 abcd	16.5 ab	5.2 a	4.7	16 ab	15	40 ab	38 a
A1830-6	15.5 def	14 cde	4.8 bcd	4.9	13 fg	14	37 abcde	33 cde
A1830-7	15 efgh	13.6 cde	4.7 cdef	4.7	14 efg	14	38 abcd	32 def
A1830-9	13.6 gh	12.9 e	4.8 bc	4.6	16 bc	14	34 efg	31 def
A1830-14	14.1 fgh	12.6 e	4.7 cdef	4.4	13 g	13	32 fg	29 f
A1830-15	13.5 h	14.1 cde	4.5 fg	4.5	14 ef	13	31 g	32 def
Local check	18.2 ab	14.9 bcd	5.2 a	4.6	17 a	14	40 a	32 def
Mean	15.9	14.5	4.7	4.5	15	13	37	33
Std	0.8	0.8	0.09	0.4	0.3	0.9	1.6	1.5
*p*-value	< 0.01	0.03	< 0.01	0.7	< 0.01	0.6	0.02	0.01
StdMSE	0.8	0.8	0.1	0.4	0.3	0.9	1.6	1.5
CV (%)	5.1	5.9	1.9	8.8	2.3	6.9	4.3	4.5

Remarks: StdMSE—standard mean sum of error, Std—standard deviation, LSD—least significant difference, CV—coefficient of variation, *p*-value < 0.01 indicates highly significance, <0.05 significant effect, >0.05—non-significant effect, similar statistical letters within a parameter (column) indicates non-significant difference among the genotypes.

## Data Availability

The data presented in this study are available on request from the corresponding author.

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
