# Peer review of "Zinc and Provitamin A Biofortified Maize Genotypes Exhibited Potent to Reduce Hidden—Hunger in Nepal"

_plants, 2022, doi:10.3390/plants11212898_

Round 1

Reviewer 1 Report

The paper identifies and characterizes Zinc and Pro- Vitamin- A Fortified Maize Genotypes to reduce hidden hunger in Nepal. The paper is ok, nothing groundbreaking but ok. I would consider uniforming the figures and add the standard deviation and statistics everywhere. Moreover, while the English is okay in some parts it sounds a bit off therefore, I would highly recommend to correct all the English by a native speaker.

A few more comments:

Intro

40 You may cite 10.3389/fpls.2017.01887

42 age group of what?

47 this sentence is wrong, rewrite it properly

49 doesn’t this depend on body mass?

55 I miss a description of “normal” Zinc/Fe (etc) content of those crops

71 isnt this a repetition of above?

Materials and methods

88-89 what is medium? Tell us the numbers

132 what is cob characteristics?

161 a few more details for zinc analysis

Figure 1 please use only English

Results

Tables. why is there no standard deviation for each value? I highly recommend to add that.

Figure 3. don’t extend the y axis to negative numbers

Fig 4. Again no standard deviation and please use one format for all figures

Fig 7 is of low quality and relatively ugly. Please improve

Author Response

Thank you very much for providing your review comments. 

Reviewer 2 Report

Dear Authors,

I have read your manuscript titled "Zinc and Pro- Vitamin- A Fortified Maize Genotypes Exhibited Potent to Reduce Hidden- Hunger in Nepal" and I found it scientifically interesting and a high quality work.

I accept this manuscript for publication after a native English revision.

Regards

Author Response

Thank you very much for providing your timely review reports.